# Optogenetically induced cellular habituation in non-neuronal cells

**Mattia Bonzanni**[1], **Nicolas Rouleau**[1], **Michael Levin**[2], **David L. Kaplan**[1] *

**1** Department of Biomedical Engineering, Allen Discovery Center, Tufts University, Medford, United States of America, **2** Department of Biology, Allen Discovery Center, Tufts University, Medford, United States of America

* david.kaplan@tufts.edu

**Data Availability Statement:** All relevant data are within the manuscript and its Supporting Information files.

**Funding:** Funded by 1) ML and DK, No. 12171, Paul G. Allen Frontiers Group, https://alleninstitute.

## Abstract

Habituation, defined as the reversible decrement of a response during repetitive stimulation, is widely established as a form of non-associative learning. Though more commonly ascribed to neural cells and systems, habituation has also been observed in single aneural cells, although evidence is limited. Considering the generalizability of the habituation process, we tested the degree to which organism-level behavioral and single cell manifestations were similar. Human embryonic kidney (HEK) cells that overexpressed an optogenetic actuator were photostimulated to test the effect of different stimulation protocols on cell responses. Depolarization induced by the photocurrent decreased successively over the stimulation protocol and the effect was reversible upon withdrawal of the stimulus. In addition to frequency- and intensity-dependent effects, the history of stimulations on the cells impacted subsequent depolarization in response to further stimulation. We identified tetra-ethylammonium (TEA)-sensitive native $K^+$ channels as one of the mediators of this habituation phenotype. Finally, we used a theoretical model of habituation to elucidate some mechanistic aspects of the habituation response. In conclusion, we affirm that habituation is a time- and state-dependent biological strategy that can be adopted also by individual non-neuronal cells in response to repetitive stimuli.

## Introduction

The behavioral manifestation of habituation is intuitive and can be simplified as a reversible asymptotic response decrement after repeated stimulations [1]. The seminal work of Thompson and Spencer [2] delineated the original characteristics of habituation which remain largely unchanged today [1]. The principal features, which are now succinctly summarized in ten points by Rankin and colleagues [1], represent the gold standard for the definition of behavioral habituation in organisms. Briefly, the habituation profile is, in most cases, an exponential-like curve and, most importantly, the decremental response is reversible–a condition that distinguishes habituation from fatigue. The dependence of the habituation profile upon the parameters of the stimulus cannot be overstated. Indeed, they are affected by both the intensity and frequency of stimulation as well as by the stimulation history (i.e., series of stimulation]. A

org/what-we-do/frontiers-group/. 2) M.L, No.
TWCF0089/AB55, Templeton World Charity
Foundation, https://www.templetonworldcharity.
org/. 3) D.K, P41EB002520, National Institutes of
Health, https://www.nih.gov/. The funders had no
role in study design, data collection and analysis,
decision to publish, or preparation of the
manuscript.

**Competing interests:** The authors have declared
that no competing interests exist.

generalizable mechanism for this phenomenon, however, is still lacking. So far, the dual process theory, proposed by Groves and Thompson [3], the stimulus-model comparator by Sokolov [4] and the "negative-image model" by Ramaswami [5] are the most prominent theories which offer explanatory value. The formulation of a general hypothesis that explains the process is challenging, mainly due to the multivariate cellular mechanisms that underlie the processes. In order to overcome this difficulty, we recently proposed a model of habituation that does not require a priori knowledge of the system's biological components [6]. Interestingly, some features of habituation can also be detected in non-neuronal system, [7] [8] [9] [10]. The evolutionary and cell-biological origins of learning are nowadays the focus of an emerging field—basal cognition; recent and classic work has sought to identify and mechanistically characterize primitive forms of learning in non-neural biological systems[11, 12]. So far, a clear understanding of the potential general nature of the habituation process has not been achieved. We took advantage of the overexpression of channelrodopsin2 (ChR2) to optogenetically stimulate human embryonic kidney (HEK) cells to highlight, if present, the fundamental similarities between behavioral and cellular manifestations of the habituation response and to potentially reveal new findings that can lead to a mechanistic understanding of the process itself. Here, we explored the first five of the ten points listed in the paper by Rankin and colleagues (as the last five points refer to special cases or instances with more than one stimulus) in the in vitro aneural system. We found that the system responded to the repetitive stimulation with a reversible asymptotical, exponential-like profile; moreover, the cell system response was stimulation-dependent. This indicates that responses associated with single non-neuronal cells share a high degree of similarity with behavioral manifestations of habituation.

## Material and methods

### Cell culture and transfection

For electrophysiological recordings, human embryonic kidney (HEK) cells were maintained in DMEM high glucose (Thermofisher) supplemented with 10% of fetal bovine serum (FBS; Gibco) and 2 mM of L-Glutamine (Sigma) at 37 C in a 5% $CO_2$ incubator. HEK were plated in 35 mm dishes and transfected with 1.5 μg of the pcDNA3.1/hChR2(H134R)-mCherry plasmid (Addgene #20938) using Lipofectamine[TM] 3000 (Thermofisher) accordingly manufacturer instructions. After 24–36 hours, mCherry-expressing cells were selected for patch clamp analysis.

### Electrophysiology and optogenetic stimulation

Patch clamp experiments in the whole-cell configuration were carried out 24–36 hours post-transfection on mCherry-expressing cells at room temperature. HEK cells were superfused with an extracellular-like solution containing (mM): NaCl 140, KCl 5.4, CaCl₂ 1.8, MgCl₂ 1, Hepes-NaOH 10, Glucose 5.5, pH = 7.4. The pipette (7–9 MΩ) were filled with an intracellular-like solution containing (mM): K-Asp 130, NaCl 10, EGTA-KOH 5, MgCl₂ 2, CaCl₂ 2, ATP (Na₂-salt) 2, creatine phosphate 5, GTP 0.1, Hepes-KOH 10; pH 7.2. Optogenetic stimulation was delivered by the OptoPatcher system using LSD-1 light stimulation device (ALA Scientific Instruments) as previously described [13]. Data acquisition and light triggering were controlled with pCLAMP software via DigiData 1440A interfaces (Molecular Devices). The channelrodopsin (ChR2) photocurrent was measured under voltage-clamp conditions from a holding potential of 0 mV applying concomitantly hyperpolarizing test steps in the range 0/-90 mV and high-intensity illumination for 2600 ms. Peak and stationary currents were normalized by cell capacitance. Patch-clamp currents were acquired with a sampling rate of 4 kHz without lowpass filter. Neither series resistance compensation nor leak or liquid junctional

potential corrections were applied. The light stimulation was delivered for 20 s as pulse train (or cosine wave) in I/0 configuration at three different frequencies (in Hz: 0.5; 1; 2) and three intensities (Low: 1V; Middle: 2V; High: 5V. Voltage values referrers to the LSD-1 light stimulation device). The mono-exponential decay fitting was used to calculate the percentage of depolarization at the steady state and the tau of habituation ($\tau_H$), defined as the number of events/time necessary to reach the 37% of the percentage of depolarization at the steady state. The probability of habituation (p(H)) was defined as 1 if the cell response fitted or 0 if the cell response did not fit with a mono-exponential fitting.

## Statistical analysis

Data were analyzed with Clampfit10 (Axon) and Origin Pro 9. To test the impact of the stimulation features on the habituation profile, we compared the mean percentage of depolarization at the steady state and the mean tau of habituation ($\tau_H$) at different conditions. These two parameters are sufficient to uniquely describe a mono-exponential profile. Data were compared using either One-Way ANOVA followed by Fisher's LSD post-hoc test or Student's T-test; significance level was set to p = 0.05. Data outliers were excluded using Tukey's method. Data were collected from different transfection experiments ranging from a minimum of four to a maximum of twelve.

# Results

## Optogenetically-induced depolarizations are reduced by repetitive stimulation

To explore the habituation process in single aneural cells, human embryonic kidney (HEK) cells were transfected with a Channelrodopsin2 (ChR2)-expressing plasmid and the functional expression of the photocurrent was assessed in mCherry-positive cells (S1 Fig). Subsequently, ChR2-expressing cells were photostimulated (pulse train) and the membrane potential ($V_{mem}$) was simultaneously recorded using a patch clamp approach in the whole cell configuration. A representative stretch of the $V_{mem}$ profile during 1Hz/5V light stimulation is shown in Fig 1A, in which the depolarization induced by the photocurrent (hv, blue lines) is visibly reduced over time. A similar reduction is also observed when the stimulation was given as cosine waves rather the pulse train (S2 Fig) suggesting the independence of the cell's response from the shape of the delivered stimulation. In the absence of the ChR2 channel expression, the light stimulation did not induce any change in the $V_{mem}$ (S3 Fig). The decremental reduction of the depolarization is summarized in Fig 1B. All data points were normalized by the magnitude of depolarization of the first event, obtaining the percentage of depolarization (y-axis, Fig 1B); data were plotted against either the number of events or time. For each profile, the percentage of depolarization at the steady state and tau of habituation (% of dep. at s.s. and $\tau_H$, respectively) are computed using a monoexponential decay fitting and used to define the magnitude (% of dep. at s.s.) and kinetic ($\tau_H$) characteristics of habituation. By definition, $\tau_H$ is the number of events or time necessary to reach 37% of the amplitude value (Fig 1B). The observed asymptotical response reduction during repetitive stimulation is a key feature necessary to define any habituation profile.

## The frequency and intensity of stimulation affects both magnitude and kinetic of habituation

The frequency and intensity characteristics of the stimulation are well-known modulators of the habituation. First, we explored the impact of the frequency of stimulation on the

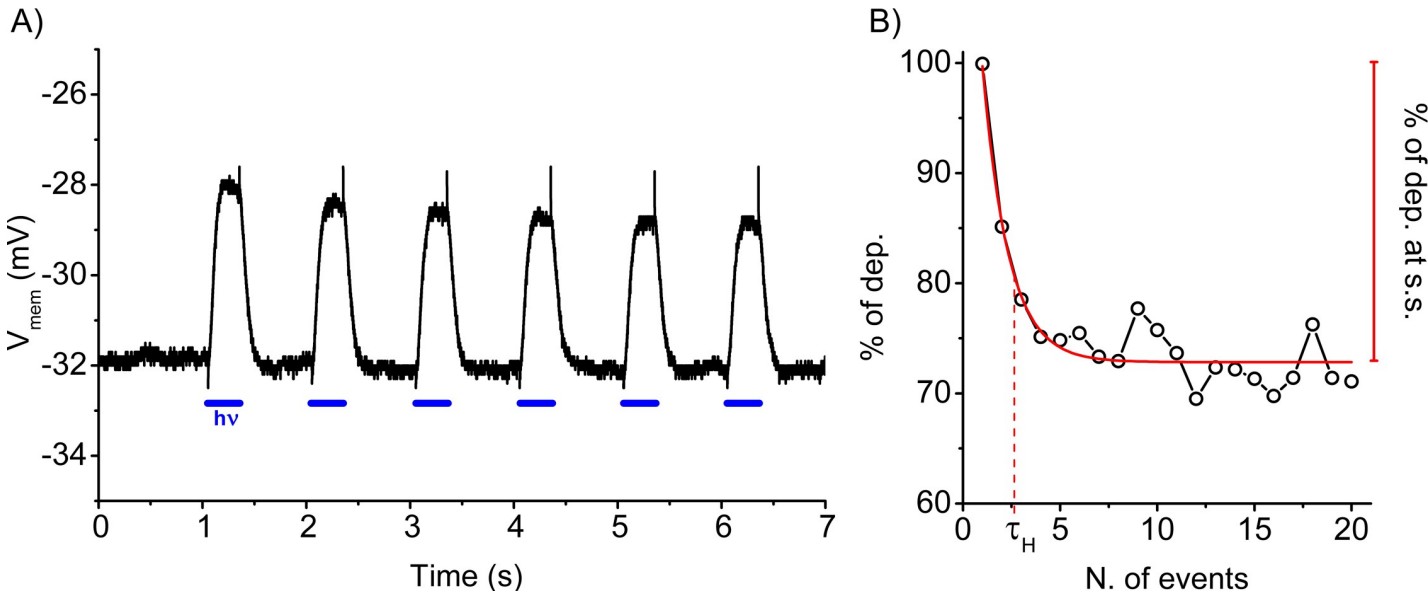

**Fig 1. Definition of the habituation profile. A)** Representative trace of voltage recorded in the I/0 configuration during a light stimulation at 465 nm (blue lines). **B)** Normalized values of depolarization during 20 s of 1Hz/5V stimulation protocol. Monoexponential fitting curve of the plotted data (circle) is shown in red. Percentage of depolarization at the steady state (% of dep. at s.s.) and tau of habituation ($\tau_H$) are indicated.

habituation profile. In Fig 2 (top panel), HEK cells were stimulated at 5V for 20 s at three different frequencies as indicated (top panel, in Hz: 0.5, 1, 2; black square, purple circle and green triangle, respectively). The resulting mean traces are shown either superimposed (Fig 2A) or divided (Fig 2B) plotting the number of events on the x-axis. Mean $\tau_H$ and % of dep. at s.s. values are summarized in Fig 2C and 2D in a frequency-dependent fashion. When we considered the number of events, other things being equal, higher stimulation frequencies were associated with a slower kinetic (Fig 2C; $p < 0.05$ among groups) and more pronounced amplitude (Fig 2D and 2H; $p < 0.05$). On the other hand, when we considered time rather than events as displayed on the x-axis (S4 Fig), higher stimulation frequency was associated with a faster kinetic (S4 Fig). From these results, the frequency of stimulation clearly affects both the kinetic and magnitude of the habituation profile indicating a frequency-dependent response.

We also explored the impact of different intensities of stimulation on the habituation profile (bottom panel). HEK cells were stimulated at 1 Hz for 20 s at three different intensities: Low: 1V; Medium:2V; High:5V (bottom panel: black square, purple circle and green triangle, respectively). The resulting mean traces were shown superimposed (Fig 2E) or separated (Fig 2F) plotting the number of events on the x-axis; mean $\tau_H$ and % of dep. at s.s. values are summarized in Fig 2G and 2H in an intensity-dependent fashion. Other factors being equal, at 1V the kinetic is significantly slower when compared to both 2V and 5V stimulations (Fig 2G). Moreover, at 1V the magnitude of habituation is less pronounced ($p < 0.05$) than both 2V and 5V conditions (Fig 2H). Taken together, these results highlight both frequency- and intensity-dependent behavior of the cellular system.

## The recovery profile is frequency-dependent

A hallmark of habituation is the reversibility of the decremental response. We thus explored the recovery profile from the steady state condition (filled symbols, Fig 3) increasing the recovery time between consecutive series of stimulations. We evaluated the recovery profile in a

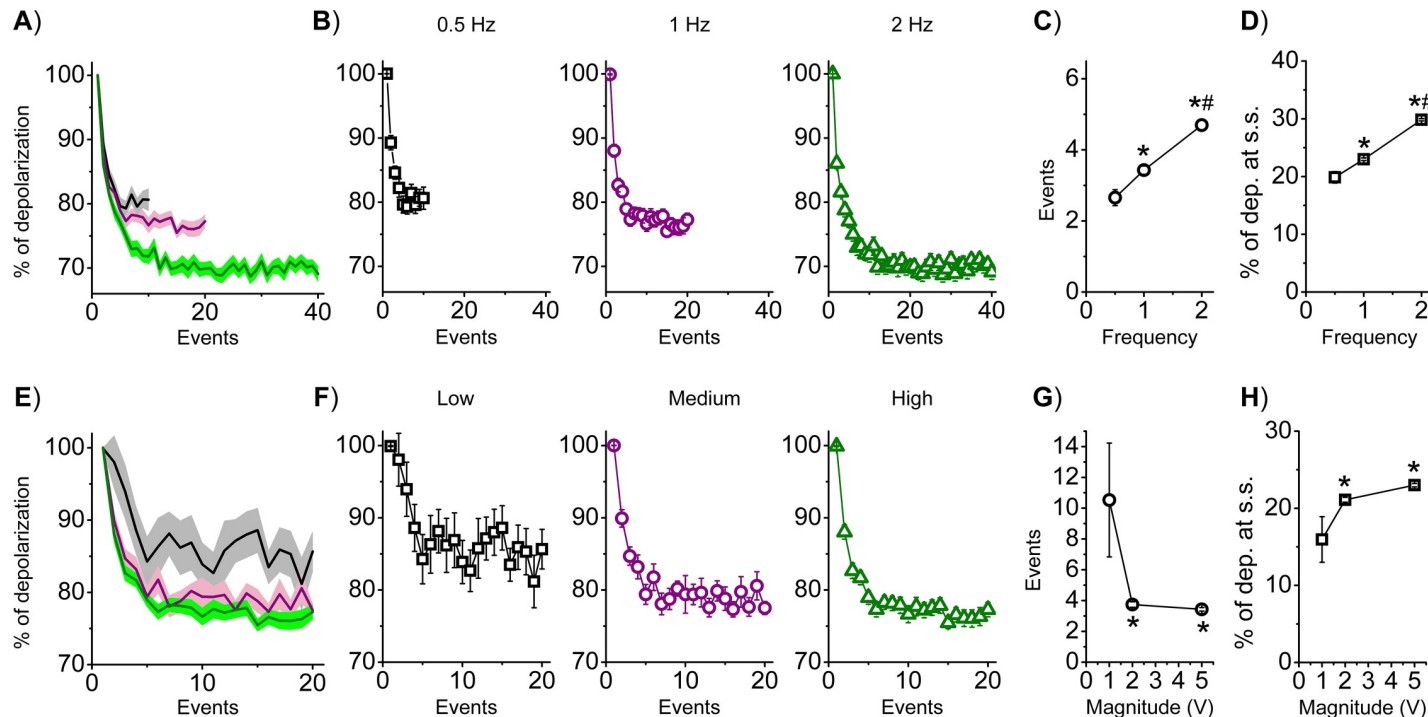

**Fig 2. The impact of the stimulation features on the habituation profile.** HEK cells were stimulated at 5V at three different frequencies as indicated (in Hz: 0.5, black square; 1, purple circle; 2 green triangle). **A)** Superimposed (solid line is the mean and colored area the S.E.M.) and **B)** separated mean profiles are shown plotting the number of events. **C)** Mean $\tau_H$ (in events: 0.5Hz: 2.66±1.00, n = 21; 1Hz: 3.43±0.12, n = 43; 2Hz: 4.70±0.16, n = 43) and **D)** mean % of dep. at s.s. (0.5Hz: 19.87±1.00, n = 21; 1Hz: 23.00±0.23, n = 43; 2Hz: 29.84±0.19, n = 43) are shown. HEK cells were also stimulated at 1Hz at three different intensities as indicated (Low: 1V black square; Medium: 2V purple circle; High: 5V green triangle). **E)** Superimposed and **F)** separated mean profiles are shown, plotting the number of events. **G)** Mean $\tau_H$ (in events: Low: 10.53±3.69, n = 12; Medium: 3.74±0.18, n = 9; High: 3.43±0.12, n = 43) and **H)** mean % of dep. at s.s. (Low: 16.00±2.96, n = 12; Medium: 21.11±0.32, n = 9; High: 23.00±0.23, n = 43) are shown in the event-domain. One-way Anova, *p<0.05 vs 0.5Hz or 1V; #p<0.05 vs 1Hz.

frequency-dependent manner. After reaching the steady state of the habituation profile, we normalized the following stimulation profile based on the first event of the first stimulation (filled symbol) and reported on the graph the mean % of depolarization after increasing recovery times (unfilled symbols). In Fig 3A, the mean recovery profiles are shown for 0.5, 1 and 2Hz (square, circle and triangle, respectively). It is clear that the time necessary to reach again the 100% of the response is frequency-independent (26.7 s). On the other hand, the recovery trajectory appeared to be conserved at 1Hz and 2Hz and different at 0.5 Hz, suggesting potential different frequency-dependent mechanisms. We then analyzed both $\tau_H$ (Fig 3B) and the % of dep. at s.s. (Fig 3C) of the profiles during the consecutive series of stimulation; the x-axis indicates the resting period between consecutive stimulations and the dotted line represent the value of the descriptor during the first stimulation (filled symbols). Both descriptors displayed a frequency signature; it is also interesting to notice that at 3.5 s and 4.2 s (1Hz stimulation) the kinetic is slower. We also reported the probability to generate a habituation profile (p(H)) (Fig 3D); we found that in all conditions, when examining cases where recovery time is below 2.3s, the probability to generate the habituation profile is null. Taken together, the results indicate that the decremental response was reversible and that, based on the recovery time, the kinetic and magnitude of the profiles have complex frequency-dependent behavior. Moreover, the probability to generate a habituation profile during consecutive stimulations is not an assumption that can be made a priori.

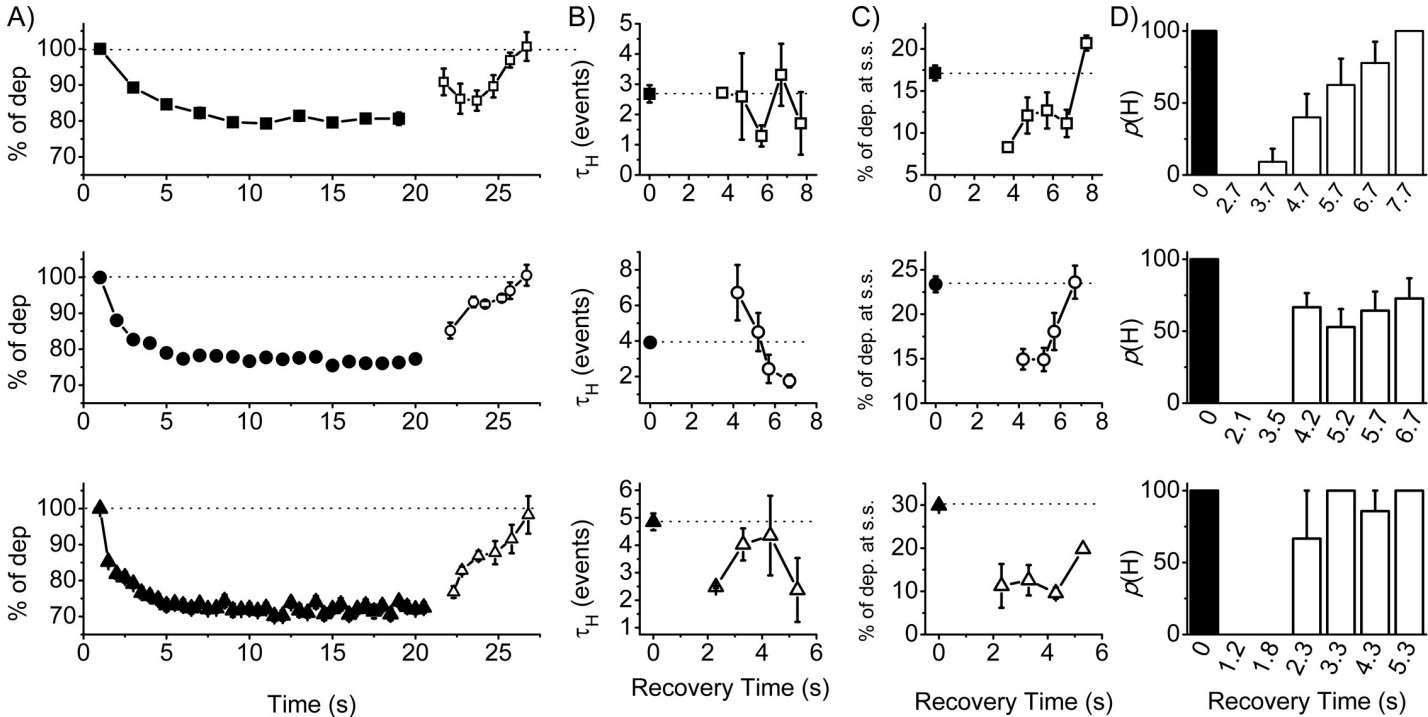

**Fig 3. Frequency-dependent recovery profile.** HEK cells were stimulated at 5V at three different frequencies (in Hz: 0.5, square, top; 1, circle, middle; 2 triangle, bottom) for 20s and, after a recovery time, the same frequency protocol was applied. **A)** Mean profiles and normalized values of the first event (filled symbols) after different resting periods (unfilled symbols). **B)** Mean $\tau_H$ (in events) and **C)** % of dep. at s.s. of the profiles at different recovery times (dot lines indicate the values of the initial profile). **D)** Mean bar graphs indicating the probability of habituation profile (p(H)) at different recovery times. Mean values are reported in S1 Table.

## Frequency transitions influence the kinetics of the habituation profile

We then explored what would happen to the cell's output if the photostimulation suddenly changed frequency without an intervening rest period. Our aim was to simulate the rhythmic transition changes that could occur in quasi-periodic biological systems. The mean profile during the 1Hz-2Hz-1Hz transition is shown in Fig 4A (1Hz purple; 2Hz green). The mean $\tau_H$ and % of dep. at s.s. values are summarized in Fig 4B and 4C, respectively. Both the kinetic profile and magnitude at 2Hz are not affected by the previous 1Hz stimulation; indeed, the values are not different from the 2Hz stimulation alone (Fig 2). However, after the 2Hz stimulation, the 1Hz profile is faster whereas the magnitude is invariant with respect to the 1Hz condition alone (Fig 2). Moreover, after the first stimulation, the change of frequency reduces the probability of generating a habituation profile to 50% (Fig 4D). The mean profile during the 2Hz-1Hz-2Hz transition is shown in Fig 4E. The first 2Hz stimulation influences the 1Hz kinetic profile during the 2Hz-1Hz transition as shown in Fig 4F; particularly, the $\tau_H$ is significantly slower compared to the 1Hz stimulation alone but, again, reached the same magnitude with a p(H) of about 60% (Fig 4H). The following 1Hz-2Hz transition did not produce any habituation profile (Fig 4H). Collectively, these results indicated that the frequency transitions without resting periods in between affect the kinetic profile but did not affect the magnitude of the habituation.

## Native channels participate in the habituation response

Since habituation and desensitization share the same decremental response over time, we analyzed the ChR2 photocurrent profile upon stimulation to address any channel-related

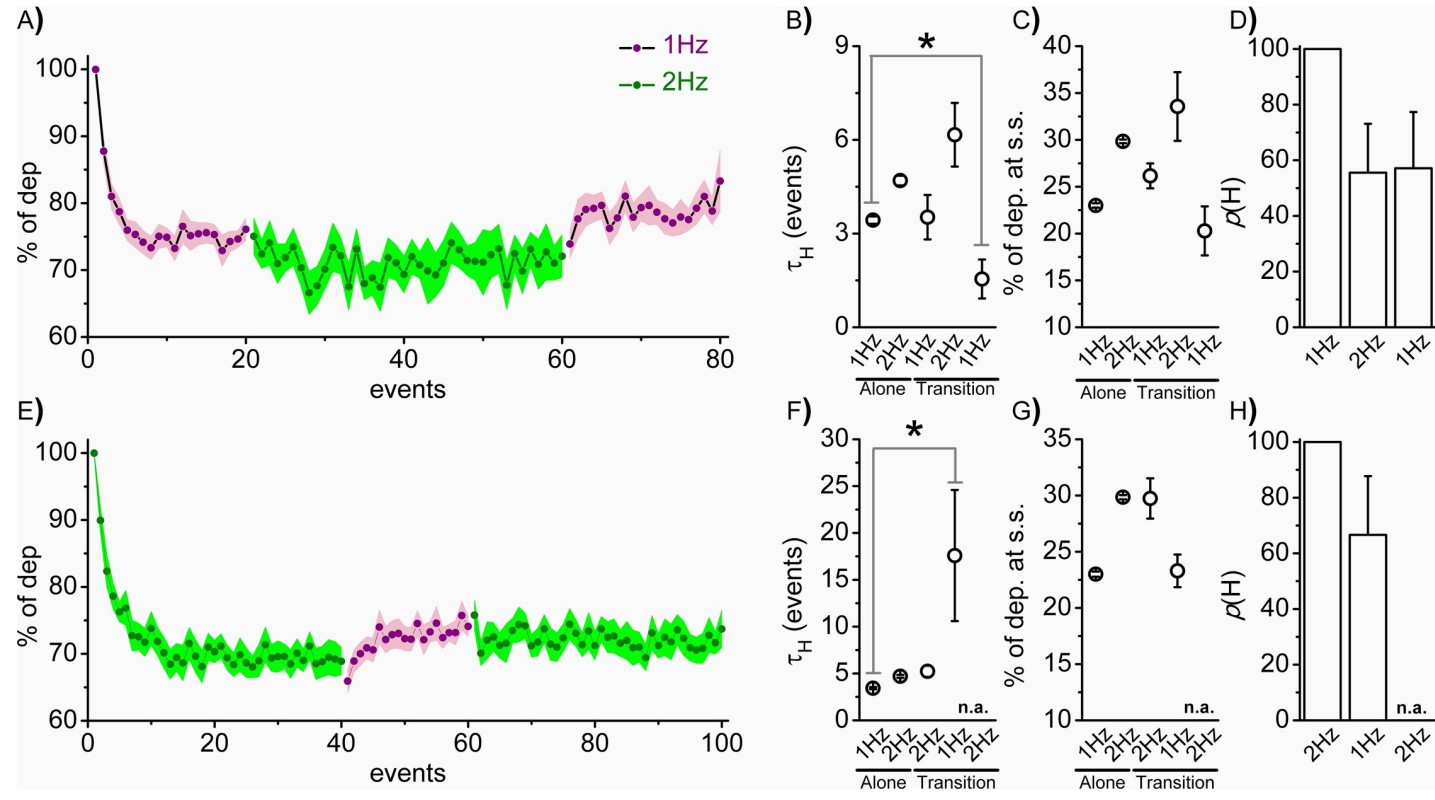

**Fig 4. Intra-protocol frequency transitions influence the habituation profile.** HEK cells were stimulated at 5V at either 1Hz (purple) or 2Hz (green) without a resting period in between. **A)** Mean profiles at 1Hz-2Hz-1Hz transition (solid line is the mean and colored area the S.E.M.). **B)** Mean $\tau_H$ (in events: Alone: 1Hz: 3.43±0.12, n = 43; 2Hz: 4.70±0.16, n = 43. Transitions: First 1Hz: 3.52±0.71; 2Hz: 6.17±1.02; Second 1Hz: 1.55±0.62, n = 12), **C)** mean % of dep. at s.s. (Alone: 1Hz: 23.00±0.23, n = 43; 2Hz: 29.84±0.19, n = 43. Transitions: First 1Hz: 26.16±1.33; 2Hz: 33.56±3.66; Second 1Hz: 20.28±2.61, n = 12) and **D)** mean bar graphs indicating the probability of habituation profile (p(H): First 1Hz: 100±0; 2Hz: 55.56±17.57; Second 1Hz: 57.14±20.20, n = 8) are shown. **E)** Mean profiles at 2Hz-1Hz-2Hz transition (solid line is the mean and colored area the S.E.M.). **F)** Mean $\tau_H$ (in events: Alone: 1Hz: 3.43±0.12, n = 43; 2Hz: 4.70±0.16, n = 43. Transitions: First 2Hz: 5.24±0.60; 1Hz: 17.59±7.0, n = 12) and **G)** mean % of dep. at s.s. (Alone: 1Hz: 23.00±0.23, n = 43; 2Hz: 29.84±0.19, n = 43. Transitions: First 2Hz: 29.73±1.79; 1Hz: 23.30±1.45, n = 12) and **H)** mean bar graphs indicating the probability of habituation profile (p(H): First 1Hz: 100±0; 2Hz: 66.67±21.08; Second 1Hz: 0, n = 12) are shown. Student's T-test *p<0.05 vs Alone condition.

desensitization effect. In Fig 5A, representative traces of the photocurrent at -30, -40 and -50 mV (square, circle and triangle, respectively) are shown during the application of the 1Hz,5V stimulation protocol for 10 seconds (blue lines); we chose three voltage values near the mean value of the resting potential of HEK cells (-40.75±1.38 mV; n = 58). The steady current was then analyzed in an event- and voltage-dependent manner. The graph in Fig 5B shows the mean density current values of the photocurrent during the applied stimulations. No significant decrement of the density current appeared during repetitive stimulation. In order to address any active cell-autonomous processes, we explored the impact of native potassium channels in the habituation process; we thus blocked them using 10 μM of TEA, as previously reported[14]. After confirming that TEA does not influence the photocurrent (Fig 5B), we analyzed the effect of the drug on the habituation profile at 1Hz, 5V. The mean profile is shown in Fig 5D and mean $\tau_H$ and % of dep. at s.s. values are summarized in Fig 5E and 5F indicating a significantly slower and more pronounced profile in the presence of TEA. This result highlights that the TEA-sensitive native potassium channels actively participate in defining the photocurrent-induced habituation process.

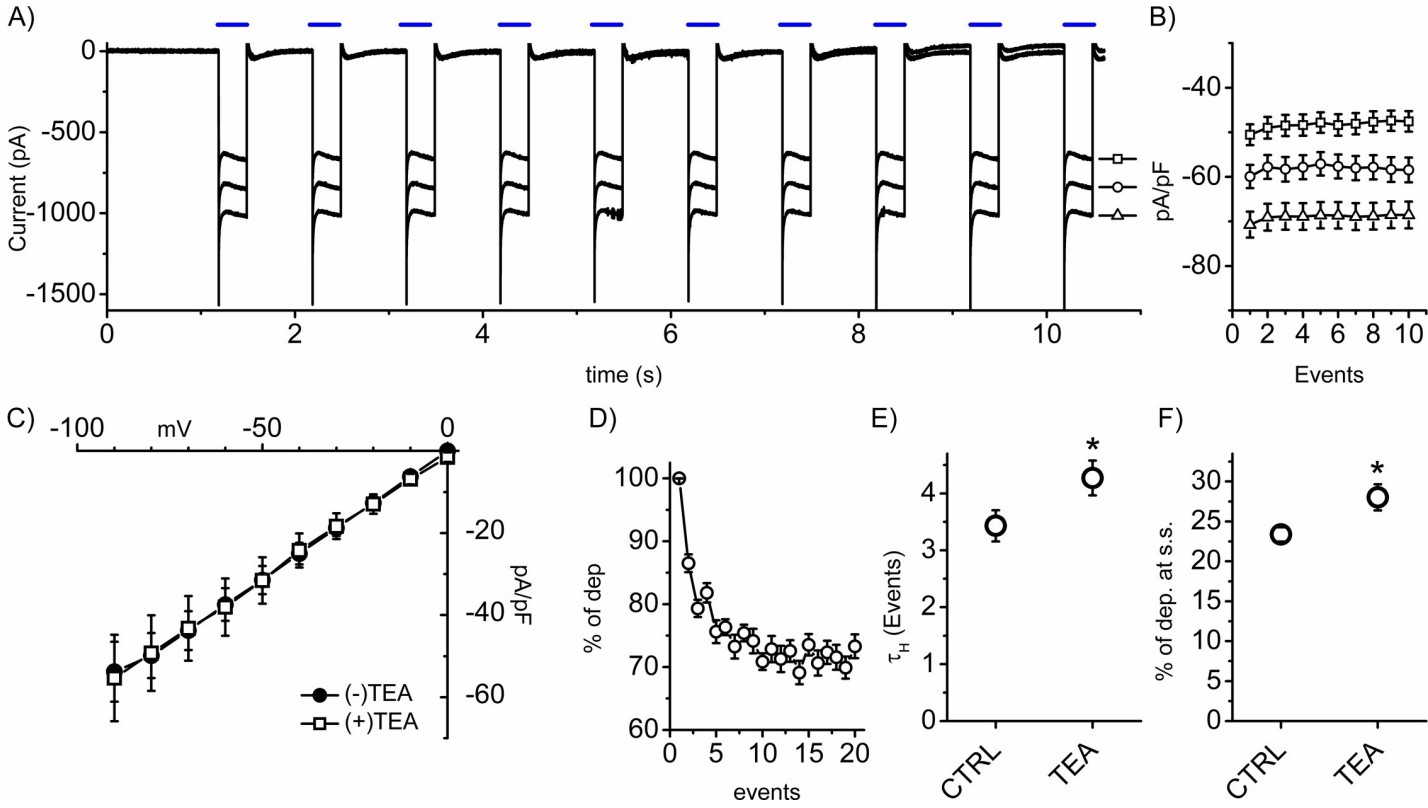

**Fig 5. ChR2-independent and ion-dependent habituation profile. A)** Representative traces of the photocurrent at -30, -40 and -50 mV (square, circle and triangle, respectively) during a 1Hz,5V repetitive stimulation. **B)** Mean current density/event plot of the photocurrent. **C)** Mean photocurrent density currents with or without TEA (filled circle, empty square, respectively; n = 8 each). **D)** Mean habituation profiles with TEA 10 µM in the extracellular solution. **E)** Mean $\tau_H$ (in events: CTRL: 3.43 ±0.27, n = 43; TEA: 4.27±0.30, n = 18) and **F)** amplitude (CTRL: 23.38±0.89, n = 43; TEA: 28.03±1.63, n = 18). Student's T-test *p<0.05 vs CTRL.

## Mathematical modeling of habituation in HEK cells

We recently proposed a generalization of the habituation process which could be applied independently of the biological details of the given system [6]. As outlined in the paper, the habituation process was described as the dynamic interplay between different elements, namely the stimulation, transducer, habituation, receiver and background elements. Each element is described by a variable and, overall, the process is described by the following equation:

$$R_n = T'_n + H'_{(ns)0} \pm \sigma \sum_{i=0}^{n-1} \Delta^i + B \tag{1}$$

where $R_n$ is the output of the receiver element (the element that we monitor during the stimulation), $T'_n$ is the output of the transducer elements (influenced by the frequency ($t_{(s)}$) and the intensity of stimulation and the nature of the modules composing the element itself), $H'_{(ns)0}$ is an index of the initial state of the habituation element and thus the output of the habituation element before the stimulation, sigma ($\sigma$) is the stimulation factor, delta ($\Delta$) is the spontaneous decay factor during the recovery phase from the stimulation, B is the output of the background elements (stimulation invariant elements) and n is the number of events delivered to the system. Through a mathematical manipulation of the Eq 1 (S1 File), we computed from the raw data $\Delta$, $\sigma$ and A (where A = $T'_n$+$H'_{(ns)0}$+B) associated with some conditions tested throughout the paper. Each parameter, as detailed in the S1 File, is influenced either by the stimulation features ($t_{(ns)}$, $t_{(s)}$ and intensity) or by the nature/composition of the habituation system (T', B

**Table 1. Relationship between the different combinations of parameters and the variables.** In the table are indicated the variables when more than one parameter is different among conditions. B is the output of the background element, H(ns)0 is the output of the habituation element before the stimulation, T' is the output of the transducer elements, int is the intensity of the stimulation, t(s) is the time of stimulation, t(ns) is the time of non-stimulation between two events and H' is the output of the habituation element.

| AND | $\Delta$ | $\sigma$ | A | $\bar{\Delta}$ | $\bar{\sigma}$ | $\bar{A}$ |
|---|---|---|---|---|---|---|
| $\sigma$ AND $A$ | All | | | B, $H_{(ns)0}$, T', int, $t_{(s)}$ | | |
| $\Delta$ AND $A$ | | All | | | B, $H_{(ns)0}$, $t_{(ns)}$ | |
| $\Delta$ AND $\sigma$ | | | All | | | H', $t_{(ns)}$ |

and $H_{(ns)0}$). The detailed relationships between the parameters ($\Delta$, $\sigma$ and A), the variables (B, $H_{(ns)0}$, T', H', $t_{(ns)}$) are reported in the S1 File.

In Table 1 we summarize the variables that can influence all the possible combinations of the significant (i.e. $\Delta$) and not significant (i.e. $\bar{\Delta}$) parameters. In Table 2 we report the significant parameters among the indicated conditions, and the variables that can be neglected are also listed (S1 File); with this eliminative procedure, we then obtained the significant variables that can explain the observed parameter combinations (for numerical details, see S1 Table). It emerges that the differences among 1Hz and 2Hz stimulations (Fig 2A) arose just from the different stimulation protocol ($t_{(ns)}$), whereas during the 0.5 Hz condition the differences must also be related to a different nature/composition of the habituation system (T', H'). When we compare 2V vs 5V (Fig 2B), we can see that a different response of T' is the explanation of the different output (in particular, reflecting the different intensities of stimulation). Upon TEA application at 1Hz 5V stimulation (Fig 5D), we can conclude that native $K^+$-channels participate either in the composition of the translator (T') or habituation element (H' and/or $H_{(ns)0}$). Finally, during the frequency transitions, the first 1Hz stimulation and the second 1Hz stimulation after the 2Hz stimulation (Fig 4A) differs because of either a difference in the pre-stimulation habituation elements ($H_{(ns)0}$) or a difference in the nature of the translator element (T'). In conclusion, the previously proposed model could be instrumental in narrowing the biological processes involved in the different responses through an experimentally-driven eliminative procedure.

## Limitations

In the present work, two main limitations are present: the non-physiological source of stimulation (the photostimulation of the ChR2) and the use of just one cellular type. Indeed, the overexpression of the ChR2 channels is an implausible physiological situation driven by the experimental need to fine-tune the stimulation features, which practically limited the use of

**Table 2. Experimental-driven eliminative procedure.** After the computation of $\Delta$, $\sigma$ and A in each group, we identified the statistically significant parameters and using Table 1 we highlight the significant variables. Moreover, based on each specific group comparison, we could also identify the variables which are invariant based on the applied stimulation.

| Experimental feature | Figure | Group Comparison | Statistically significant parameters | Neglectable Variables | Significant Variables |
|---|---|---|---|---|---|
| Frequency | Fig 2A | 0.5 *vs* 1 Hz | $\Delta$ AND $A$ AND $\sigma$ | B, mag, $H_{(ns)0}$, $t_{(s)}$ | H', T', $t_{(ns)}$ |
| | Fig 2A | 0.5 *vs* 2 Hz | $\Delta$ AND $A$ AND $\sigma$ | B, mag, $H_{(ns)0}$ | H', T', $t_{(ns)}$, $t_{(s)}$ |
| | Fig 2A | 1 *vs* 2 Hz | $\Delta$ AND $A$ | B, mag, $H_{(ns)0}$ | $t_{(ns)}$ |
| Intensity | Fig 2E | 2 *vs* 5 V | $A$ AND $\sigma$ | B, $H_{(ns)0}$, $t_{(s)}$, $t_{(ns)}$ | T', mag |
| Native Channels | Fig 5D | (-)TEA vs (+)TEA | $\Delta$ AND $A$ AND $\sigma$ | B, mag, $t_{(s)}$, $t_{(ns)}$ | T', H', $H_{(ns)0}$ |
| Frequency transitions | Fig 4A | First *vs* Second 1Hz | $A$ AND $\sigma$ | B, mag, $t_{(s)}$, $t_{(ns)}$ | T', $H_{(ns)0}$ |

more biologically relevant stimulation sources. On the other hand, the ionic currents generated by the opening of the channels (from which the depolarization arose) is a universal language for cells. Nonetheless, it is important to mention that the use of a single type of channelrodopsin prevents us to conclude which ionic current-dependent phenomena (namely the depolarization of the membrane or any other ion-dependent mechanisms) is responsible for the habituation. Moreover, we explored the process only in HEK cells (since it is a well-established heterologous system in electrophysiology); this limits any robust claim of generalization of the presented results to other non-neuronal system. Finally, we only explored the non-associative aspect of the habituation, namely using one and only one form of stimulation and, because of the intrinsic instability of the whole cell configuration over long recording periods (more than an hour), we did not explore any potential long-term effects of the stimulation. In light of these limitations, the present work should be seen as a proof of concept of the ability of non-neuronal cells to habituate rather than an indication for habituation as a biologically universal process with defined features and rules; more data must be collected to prove this claim.

## Discussion

Whether they are self-generated by the body (i.e. heartbeat, brain waves, circadian rhythms, hormone release, etc.) or delivered from environmental sources (new drug treatment, training, routine behaviors, etc.), repetitive stimulations are ubiquitous and essential to the adaptive behavior and physiology of living organisms. A common behavioral strategy to deal with repetitive stimulations is to reversibly reduce the output of the system; a process which is termed habituation [2]. Over the last 50 years, an extensive characterization of the behavioral manifestation of habituation has been performed [1] mostly confirming the characteristics previously identified [2]. So far, the list of features reported by Rankin and colleagues [1] represents the most up-to-date guideline to correctly classify behavioral habituation. Habituation is considered within an exclusively neural-based framework even though some experiments demonstrate the process clearly emerges within aneural systems [7] [8] [9] [10]. While data continue to accumulate to broaden our view of the gradual evolution of learning capacities from basal taxa, it is essential to develop platforms that facilitate the study of universal cellular mechanisms for computation and optimization of behavior. While single-cell habituation is apparently robust, a deeper characterization has not yet been achieved. A proper comparison between the cellular and behavioral manifestations of habituation could reveal a more general process that is not restricted to neuronal substrates.

In the present work, we explored the habituation process in ChR2-expressing HEK cells. The main advantage of using the ChR2 is to uniquely stimulate a singular element of the cell (indeed, the blue light stimulation did not affect the resting membrane potential of the cells, the output that we monitored throughout the study). The impact of ChR2-mediated depolarization on the voltage profile of the cells was studied, defining three descriptors: percentage of depolarization at the steady state (% of dep. at s.s.) and $\tau_H$ to describe the magnitude and kinetic of habituation, respectively, and p(H), the probability to generate an exponential-like profile. From Fig 1A, the repetitive series of stimulations decreased the amplitude of the photo-current-induced depolarization within the protocol with an asymptotic profile (Fig 1B). It is also important to notice that the photocurrent amplitude was invariant throughout the stimulation (Fig 5B), demonstrating that the decrement was ChR2 independent. In support of this hypothesis, the blockage of native potassium channels with TEA changed the profile's features (Fig 5D) indicating that the cell was actively responding to the repetitive stimulation; it is also relevant to mention that TEA dosage did not influence the photocurrent characteristics (Fig

5C). Finally, the recovery of the output after a resting period (Fig 3) led us to exclude any deleterious effects of the stimulation on the cell output. Taken together, these results point toward a robust indication of habituation in the analyzed cell system.

As previously described from a behavioral standpoint [1], the stimulation characteristics must affect the response. We thus tested the impact of different frequencies of stimulation (Fig 2) finding that increasing the stimulation frequency produced a more pronounced profile (Fig 2D). Plotting the time on the x-axis, higher stimulation frequencies were associated with faster profiles (S4 Fig), which is in line with the behavioral data; we observed the opposite effect when plotting the number of events (Fig 2C). This apparent contradiction highlights the necessity to always clarify if the analysis of the kinetic is made with respect to either time or events. We then manipulated the intensity of the stimulation and found a less pronounced and slower profile at 1V ($p < 0.05$) and no differences between 2V and 5V. Taking into consideration the limited range of intensities that we explored, our results are clearly in opposition with the behavioral data. So far, we discussed the response of the cell system to a novel stimulation; in Fig 3, however, we explored the profile after consecutive stimulations. We found that the kinetic profile was not necessarily faster after consecutive stimulations, as it was framed for the behavioral habituation; a similar contradiction between behavioral and cellular data can also be highlighted when considering the magnitude. Most importantly, it emerged that habituation cannot be considered granted without satisfying certain temporal criteria; indeed, below a recovery period of 2.3 s, it seems that the cells cannot generate any habituation profile (Fig 3D). An absolute habituation refractory period emerged below which the habituation itself could not occur; in other words, the habituation elements in the system are not responsive during the absolute habituation refractory period.

Moreover, in Fig 4 we explored systematic changes in rhythmicity without deliberate recovery. This protocol was designed to mimic physiological changes in the frequency of biological periodic stimulation: actually, considering stimulations that arise inside the body, it is more common that the system experiences a modification in the rhythmic event rather than a new type of stimulation. It appears that the kinetic, but not the magnitude, was affected by the sequence of the frequency transitions. It follows that the magnitude of habituation can be considered the only invariant frequency-dependent signature during the frequency transitions. Most importantly perhaps, it highlights that the same stimulation (1Hz) can lead to either a habituation or sensitization profile based on the pre-1Hz stimulation state (Novel vs 2Hz vs 1Hz-2Hz). The evidence that habituation and sensitization arise from the same protocol of stimulation suggests that the state of the system before the stimulation is a crucial factor, more so than the features of the stimulation itself in defining the ultimate phenotype. In particular, we can speculate that a habituation profile emerges if the % of dep. at s.s. of the previous state is smaller than the one associated with the frequency of the second stimulation; on the other hand, if it is greater, a sensitization profile emerges. It also leads to the speculation that habituation and sensitization are two facets of the same process. In other words, the system seems to achieve a defined frequency-dependent steady state using either habituation or sensitization phenomena accordingly to the previous state of the system. The determinant of whether one emerges over the other would be the pre-stimulation state of the system; however, any robust conclusion cannot be irrefutable considering the limited data presented here. Most importantly, perhaps, this establishes the experimental basis to explore the effect, if any, of pathophysiological changes of rhythmic processes generated by excitable cells (i.e. cardiomyocytes, neurons) on non-excitable cells (i.e. endothelial cells, fibroblasts, macrophages, microglia). Even if we confirmed the habituation process in HEK cells, those results reveal little about any mechanistic explanation.

Using a mathematical generalization of the habituation process [6], we narrowed some potential mechanisms of the habituation in the present system. In particular, we can see that any difference between 1Hz and 2Hz is due to just the different frequency but not because of recruitment/dismissal of elements in the HEK system: in other words, the system that reacts to the stimulation is, in both activity and composition, identical. A similar picture arises when we compared 2V vs 5V. On the contrary, when we analyzed the 0.5Hz vs 1Hz (or 2Hz) stimulation, we realized that the differences were not only because of the different stimulation protocols, but also because of a different activity/composition of the HEK system reacting at those frequencies. In other words, different frequencies are processed differently by the system because of a change in its state. This hypothesis also seems to be reflected in the different profile of the recovery in Fig 3.

Taken together, our data show that both the behavioral and our cellular model share a decremental decrease during repetitive stimulation that, after a resting period, is reversible. Moreover, they both showed a frequency and intensity dependence of the habituation profile; however, it is critical to report that the similar changes in the stimulation features do not necessarily lead to the same habituation profile changes in the behavioral vs cellular comparison. The authors suggest that this is due to the fact that the specific response to stimulation changes are not amenable to generalization. Namely, the responses lie on the peculiar composition of the system that we are monitoring and must be tested de novo for any new system. To summarize, the behavioral and cellular habituation processes shares 1) an asymptotical decrement of the output during repetitive stimulation, 2) the reversibility of the profile after a resting period and 3) a dependence on both frequency and intensity of stimulation. Based on these findings, we propose to consider and define habituation as a time- and state-dependent process which could occur if and only if 1) the time between two consecutive stimulations is smaller than the time necessary to the system to achieve a pre-stimulation state and larger than the absolute habituation refractory period, 2) satisfy the three points above-mentioned. Future experiments using many more cell substrates will test the solidity of our definition and clarify any claim as to the universality of the habituation process.

## Conclusions

Bearing in mind the aforementioned limitations, the present work: 1) demonstrates that non-neuronal cells can habituate in a stimulation-dependent manner, 2) highlights similarity and discrepancies between the behavioral rules and our model responses, 3) gives defined descriptors to analyze the process (% of effect at s.s., $\tau_H$ and probability of habituation), 4) shows that systems respond differently in case of preceding history of stimulation and 5) guides the exploration of mechanistic information using an experimental-driven shortcut approach based on a mathematical generalization of the habituation process.

## Supporting information

**S1 File. Description of the mathematical model.**
(DOCX)

**S1 Fig. Photocurrent current density-voltage plot. A)** Representative photocurrent density traces (holding potential: 0 mV) recorded in the range 0/-90 mV ($\Delta V$ = 10 mV). **B)** Current density-voltage plot analyzed at the peak (square) or steady state (circle). n = 14.
(TIF)

**S2 Fig. Cosine wave-induced habituation profile. A)** Representative voltage trace upon the application of **B)** a 1Hz,5V cosine wave light stimulation.
(TIF)

**S3 Fig. Non-transfected HEK cell does not respond to light.** Representative voltage profile of non-transfected HEK cells (in black) in response to the light stimulation protocol (in blue).
(TIF)

**S4 Fig. The stimulation's features impact the habituation profile.** HEK cells were stimulated at 5V at three different frequencies as indicated (in Hz: 0.5, black square; 1, purple circle; 2 green triangle). **A)** Superimposed and **B)** separated mean profiles are shown plotting the time pf stimulation. **C)** Mean $\tau$H (in s: 0.5Hz: 6.11±0.81, n = 21; 1Hz: 3.43±0.12, n = 43; 2Hz: 2.32 ±0.11, n = 43) and **D)** mean amplitude (in % of depolarization: 0.5Hz: 19.78±1.00, n = 21; 1Hz: 23.00±0.23, n = 43; 2Hz: 29.84±0.19, n = 43) are shown. One-way Anova *p<0.05 vs 0.5Hz; #p<0.05 vs 1Hz.
(TIF)

**S1 Table. Fig 3 parameters details.**
(TIF)

## Author Contributions

**Conceptualization:** Mattia Bonzanni, Nicolas Rouleau.

**Data curation:** Mattia Bonzanni.

**Formal analysis:** Mattia Bonzanni.

**Funding acquisition:** Michael Levin, David L. Kaplan.

**Investigation:** Mattia Bonzanni.

**Methodology:** Mattia Bonzanni.

**Project administration:** Mattia Bonzanni.

**Resources:** Michael Levin, David L. Kaplan.

**Supervision:** Mattia Bonzanni.

**Validation:** Mattia Bonzanni.

**Visualization:** Mattia Bonzanni.

**Writing – original draft:** Mattia Bonzanni, Nicolas Rouleau.

**Writing – review & editing:** Mattia Bonzanni, Nicolas Rouleau, Michael Levin, David L. Kaplan.

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
