## [Decision Letter · Decision Letter 0]

2 Dec 2019

PONE-D-19-25035

Optogenetically induced cellular habituation in non-neuronal cells

PLOS ONE

Dear Dr. Kaplan,

Thank you for submitting your manuscript to PLOS ONE. After careful consideration, we feel that it has merit but does not fully meet PLOS ONE’s publication criteria as it currently stands. Therefore, we invite you to submit a revised version of the manuscript that addresses the points raised during the review process.

We would appreciate receiving your revised manuscript by Jan 16 2020 11:59PM. To enhance the reproducibility of your results, we recommend that if applicable you deposit your laboratory protocols in protocols.io, where a protocol can be assigned its own identifier (DOI) such that it can be cited independently in the future. For instructions see: http://journals.plos.org/plosone/s/submission-guidelines#loc-laboratory-protocols

We look forward to receiving your revised manuscript.

Kind regards,

Mark S. Shapiro

Academic Editor

PLOS ONE

Journal requirements;

1. We note that you have included the phrase “data not shown” in your manuscript. Unfortunately, this does not meet our data sharing requirements. PLOS does not permit references to inaccessible data. We require that authors provide all relevant data within the paper, Supporting Information files, or in an acceptable, public repository. Please add a citation to support this phrase or upload the data that corresponds with these findings to a stable repository (such as Figshare or Dryad) and provide and URLs, DOIs, or accession numbers that may be used to access these data. Or, if the data are not a core part of the research being presented in your study, we ask that you remove the phrase that refers to these data.

Additional Editor Comments (if provided):

The journal apologizes for the extreme tardiness in the review; however, we had substantial difficulties in finding reviewers for this manuscript. The overall decision was based on the opinion of an expert reviewer and the reading of the manuscript by the academic editor. I would very strongly recommend following the recommendations of the reviewer, if the authors decide to re-submit to the journal.

Reviewers' comments:

Reviewer's Responses to Questions

**Comments to the Author**

1. Is the manuscript technically sound, and do the data support the conclusions?

Reviewer #1: Partly

2. Has the statistical analysis been performed appropriately and rigorously? 

Reviewer #1: I Don't Know

3. Have the authors made all data underlying the findings in their manuscript fully available?

Reviewer #1: Yes

4. Is the manuscript presented in an intelligible fashion and written in standard English?

Reviewer #1: Yes

5. Review Comments to the Author

Reviewer #1: A submitted manuscript on “Optogenetically induced cellular habituation in non-neuronal cells” is an interesting and important research work in this area. Authors have presented a meaningful research finding related to optogenetic modulation of the ionic current in a non-neuronal cells and its affect on habituation response. I enjoyed reading this research with lots of information and, nicely written and presented.

However, It would be great to repeat key experiments of the habituation aspects with the stable cell lines of HEK expressing ChR variant. In order to visualize full “ Global” cellular response of the optogenetic modulated ionic current on habituation in the non-neuronal cells, authors may consider to perform comparative transcriptomic/ proteomics analysis. This might give a clue about the effect of optogenetically modulated ionic current associated with unknown gene regulation and/ or protein-protein interactions/ protein levels.

6. PLOS authors have the option to publish the peer review history of their article (what does this mean?). If published, this will include your full peer review and any attached files.

Reviewer #1: No

---

## [Author Response · Author response to Decision Letter 0]

7 Dec 2019

Response to Reviewer Inputs:

1) “We note that you have included the phrase “data not shown” in your manuscript. Unfortunately, this does not meet our data sharing requirements.”

Accordingly, we edited the manuscript. In particular, we added a supplementary figure (Suppl. Fig.3) and removed the sentence. 

1) “Has the statistical analysis been performed appropriately and rigorously? I do not know”

In order to clarify the performed statistical analysis, we added the following sentences (in red) in the “Statistical Analysis” section:

“Data were analyzed with Clampfit10 (Axon) and Origin Pro 9. To test the impact of the stimulation features on the habituation profile, we compared the mean percentage of depolarization at the steady state and the mean tau of habituation (τH) at different conditions. These two parameters are sufficient to uniquely describe a mono-exponential profile. Data were compared using either One-Way ANOVA followed by Fisher's LSD post-hoc test or Student’s T-test; significance level was set to p=0.05. Data outliers were excluded using Tukey's method. Data were collected from different transfection experiments ranging from a minimum of four to a maximum of twelve.”

2) “However, it would be great to repeat key experiments of the habituation aspects with the stable cell lines of HEK expressing ChR2 variant. In order to visualize full “Global” cellular response of the optogenetic modulated ionic current on habituation in the non-neuronal cells, authors may consider to perform comparative transcriptomic/ proteomics analysis. This might give a clue about the effect of optogenetically modulated ionic current associated with unknown gene regulation and/ or protein-protein interactions/ protein levels.”

We thank the reviewer for the comment; as correctly pointed out, our data do not explore the molecular response of the system. Even if it appears to be a limitation in the manuscript, we purposely omitted any molecular investigations. Indeed, the primary aim of the project was to test phenotypic rather than mechanistic similarities between organism-level behavioral and single cell manifestations of habituation. Also at organism level, the molecular pathways are multiple and rarely generalize. As we suggested in the Discussion section, habituation, similarly to positive/negative feedbacks, must be considered a biological strategy to deal with repetitive stimulation and thus studied at the phenotypic level. Even if the reviewer’s suggestions are appropriate, we do feel that they do not apply in the presented cellular model. In this light, we would like to explain our reasons:

• Conceptual perspective: As we verbalized in the Discussion section, even if not fully supported by our limited dataset, we do believe that habituations systems do not necessarily share mechanistic processes. It follows that the significance of any molecular insights will be limited to this unique and non-physiological system. It follows that the transcriptomic/proteomics analysis will lack translatability to other systems. As we pointed out in the Limitation Section, the choice of the presented model was dictated by methodological rather than biological considerations. We thus feel that any molecular insight could not be generalized to more relevant systems. 

• Technical perspectives: Considering the fast onset of the habituation effect (<2 seconds), we are confident that the main effect is due to the modulation of the activity and/or recruitment of the existing repertoire of proteins already present at the time of stimulation. Similarly, in neuronal substrates the short term learning effects are due to either cytosolic events, such as increase of cAMP and phosphorylation events, or the insertion in the membrane of pre-existing pools of proteins (such as AMPA receptors in the early events of LTP). Transcriptional and protein changes occur later and are also associated with a different phenotypic response (short- vs long-term effects).

---

## [Editor Report · Decision Letter 1]

16 Dec 2019

Optogenetically induced cellular habituation in non-neuronal cells

PONE-D-19-25035R1

Dear Dr. Kaplan,

We are pleased to inform you that your manuscript has been judged scientifically suitable for publication and will be formally accepted for publication once it complies with all outstanding technical requirements.

With kind regards,

Mark S. Shapiro

Academic Editor

PLOS ONE
---

## [Editor Report · Acceptance letter]

10 Jan 2020

PONE-D-19-25035R1 

Optogenetically induced cellular habituation in non-neuronal cells 

Dear Dr. Kaplan:

I am pleased to inform you that your manuscript has been deemed suitable for publication in PLOS ONE. Congratulations! Your manuscript is now with our production department. 

With kind regards,

on behalf of

Dr. Mark S. Shapiro 

Academic Editor

PLOS ONE